# Predicting Food Consumption to Reduce the Risk of Food Insecurity in Kazakhstan

Aigerim Duisenbekova [1,*], Monika Kulisz [2], Alina Danilowska [3], Arkadiusz Gola [4,*] and Madina Ryspekova [1]

1  Department of Economics and Entrepreneurship, L.N. Gumilyov Eurasian National University, 010000 Astana, Kazakhstan; madina10081957@mail.ru
2  Faculty of Management, Lublin University of Technology, 20-618 Lublin, Poland; m.kulisz@pollub.pl
3  Institute of Economics and Finance, Warsaw University of Life Sciences, 02-787 Warszawa, Poland; alina_danilowska@sggw.edu.pl
4  Faculty of Mechanical Engineering, Lublin University of Technology, 20-618 Lublin, Poland
*  Correspondence: aigerim.duisenbekova95@gmail.com (A.D.); a.gola@pollub.pl (A.G.)

**Abstract:** In modern times, the risk of food insecurity is a concern for policymakers at the global and national levels, as the issue of hunger and malnutrition still exists. Food security is vulnerable to any crises. The main goal of this paper is to create a neural-network-based predictive model to forecast food consumption trends in Kazakhstan, aiming to reduce the risk of food insecurity. The initial phase of this study involved identifying socioeconomic factors that significantly influence food consumption behaviors in Kazakhstan. Principal component analysis was used to identify key variables, which became the basis for modelling artificial neural networks. It was revealed that the poverty rate, GDP per capita, and food price index are pivotal determinants of food consumption in Kazakhstan. Two models were prepared: to predict food consumption on a national scale per capita per month, and to predict the percentage distribution of various food categories. The prediction of the percentage distribution of various food categories in Kazakhstan demonstrates the positive modelling quality indicators and strengthens the assumption that network modelling can be used. Predictions for total food consumption over the next three years indicate declining metrics, raising concerns about the potential food insecurity risk in Kazakhstan.

**Keywords:** food insecurity; food consumption; data analysis; modelling; neural network; principal component analysis

## 1. Introduction

A crucial part of human activity throughout history has been related to providing the right amount of food. Although humanity has made significant progress in this regard in contemporary times, the concept of food security identified at the World Food Conference in Rome in 1974 and developed at World Food Summits in 1996 and 2009 still receives considerable attention from policymakers at the global and national levels, as hunger and malnutrition have been not eliminated. The fundamental idea behind food security is that all individuals have economic, social, and physical access to an adequate and nutritious food supply to meet their dietary needs for a stable and active life (FAO 2009). Such an approach is demanding and ambitious and, in its current formulation, can be considered as a long-term objective for developing and developed countries alike. In the contemporary context, the focus is on the nutritional dimension of food security. In 2015, United Nations member states set up the 2030 Agenda for Sustainable Development, with 17 Sustainable Development Goals (United Nations 2015). Goal 2 is about creating a world free of hunger by 2030. Achieving this goal appears challenging in light of the FAO's indicator reflecting the overall level of food security: the prevalence of undernourishment. Globally, this indicator increased from 7.5% in 2017 (the lowest level in the history of the assessment)

to 7.9% in 2019 (pre-pandemic) and 8.9% in 2020. The upward trend persisted in 2021, reaching 9.3%, and in 2022 it remained relatively unchanged (FAO 2023b).

However, this issue varies spatially. In highly developed countries, food insecurity occurs, but the problem is rather distributive; however, it is very harmful to the affected groups. For example, in the United States in 2021, 3.8% (5.1 million) of U.S. households had deficient food security (Coleman-Jensen et al. 2022). Developing countries suffer from insufficient amounts of food. Hunger and undernourishment are serious problems, particularly in numerous Asian and African countries. According to the FAO (2023b), hunger affected 19.7% of the African population and 8.5% of the Asian population in 2021. Zereyesus et al. (2022) noted with concern that the number of food-insecure people in 2022 in the 77 low- and middle-income countries covered by their investigation had increased by 9.8% (118.7 million people) from the 2021 estimate. However, this indicator remained relatively low (at 2.5%) in Europe and Central Asia over the past few years. Despite this, 116.3 million people experienced moderate or severe food insecurity in the region, and over the past two years this number increased significantly by 25.5 million people. In Central Asia, 20.2% of the population faced moderate or severe food insecurity (FAO 2023a).

There has been substantial existing research on food security, both globally (Godfray et al. 2010; Stephens et al. 2018; Lal 2016; Saboori et al. 2022) and within specific countries (Lv et al. 2022; Islam 2014; Loginov 2020; Wineman 2016). However, there are notable gaps in our understanding of the specific dynamics of food consumption patterns, particularly within the framework of developing economies such as Kazakhstan. Studies on food consumption in Kazakhstan have predominantly concentrated on specific categories like livestock (Liang et al. 2020; Akhmetova et al. 2022) or fruits and vegetables (Abe et al. 2013), with limited attention to overall food consumption. However, there is a lack of recent research, often missing detailed examinations of socioeconomic variables affecting food consumption, especially in the aftermath of geopolitical conflicts and economic shocks. For instance, previous research has primarily focused on macro-level factors such as agricultural production (Gizzatova et al. 2014) and imports (Denissova and Rakhimberdinova 2021), often neglecting consumer-related factors. The recent global events have underscored the necessity for targeted research on the relationships between socioeconomic factors and food security in the region.

Food security is especially vulnerable to risks stemming from shocks in the economy or its environment. Over the period 2000–2022, three major shocks occurred: the global financial crisis of 2008–2009 (resulting in the most significant and sharpest decline in global economic activity of the modern era), the global pandemic of COVID-19, and the Ukrainian–Russian war. The modes and strengths of the influence of each shock differ. According to Travasso et al. (2023), all over the world, the 2008 crisis drove 97 million more people compared to the pre-crisis year 2007 into hunger. COVID-19 was more severe, driving 112.6 million more into hunger. Due to the conflict in Ukraine, the number of people who are food-insecure or at high risk has soared to reach a record high of 345 million. Countries that are highly dependent on food imports are particularly susceptible to food insecurity because of this conflict (Abay et al. 2023). Approximately 50 million people across 45 countries are at risk of falling into famine or famine-like conditions (WFP 2023a). Jarrell et al. (2023) noted that despite international efforts successfully slowing down the hunger crisis in 2008–2009, they did not establish a support system for food security that would prevent the magnitude of today's (2023 perspective) crisis. This means that neither international organizations nor particular countries have implemented appropriate measures to mitigate the risk of food insecurity stemming from different shocks and create secure, sustainable, and resilient food systems.

The volume and structure of food consumption per capita are the primary indicators of meeting the nutritional needs of individuals. Ensuring food security is associated with two key indicators: the production and consumption of food products (Rabbi et al. 2021). These are determined by complex factors: economic, political, demographic, and social. The shocks mentioned above can drastically change some factors while leaving others

unaffected. At the macro level, regulatory bodies such as governments and state agencies seek to understand their power and direction of impact to encourage structural changes in the food and agriculture industries or respond to challenges (USDA ERS 2023). The European Food and Safety Authority emphasizes that consumption data are essential for assessing how people are exposed to potential risks in the food chain (EFSA 2023).

The problem of undernourishment in Kazakhstan was significant in the early 2000s. The undernourishment rate was 6.5%, and the trend was upward. Since 2009, the situation has gradually improved, but the problem still requires a satisfactory resolution. The average rate of moderate or severe food insecurity was 2.1% for 2017–2019, but it increased to 2.7% for 2019–2021, reflecting the impact of the COVID-19 pandemic shock (FAO 2023a). This is why food security is a paramount goal in agricultural and economic policies in Kazakhstan (Aimurzina et al. 2018), becoming one of the principal conditions for ensuring national security.

Despite the absence of specific legislation directly addressing food security (Aigarinova et al. 2014), the Act "About National Security of the Republic of Kazakhstan" recognizes its importance by defining food security as a state of the economy, including the agro-production sector, wherein the government can ensure the physical and economic access of the nation to quality and secure food products, sufficient to meet the physiological consumption rates and accommodate demographic growth (Parliament of the Republic of Kazakhstan 2023). "The National Plan for Ensuring Food Security of the Republic of Kazakhstan for the Years 2022–2024" (Republic of Kazakhstan 2022) outlines a comprehensive action strategy encompassing the development of agricultural production, advancement of agrarian science, education, and knowledge dissemination within the agricultural sector, ensuring economic access to food products, and enhancing the quality and safety of food products. Projected outcomes of these steps include the ability to forecast prices for the most notable food products for 1.5–2 months, proposing targeted support for socially vulnerable segments of the population, and assessing the feasibility of establishing stabilization funds. Furthermore, the uneven distribution of the risk of food insecurity among social groups exacerbates the challenge of adequately directed support, as indicated by the FAO, WFP, and other sources (Rabbitt et al. 2017; Pool and Dooris 2022; Grimaccia and Naccarato 2022; WFP 2023b; FAO 2023c).

Following the unexpected Russian invasion of Ukraine, which has noticeably impacted Kazakhstan's economy (Agaidarov et al. 2023), achieving the objectives outlined in the National Plan has been made challenging. As Kazakhstan is a close economic partner of Russia, internal and external shocks related to Russia, especially sanction measures, have a negative economic effect on Kazakhstan. For instance, rapidly rising inflation poses a serious threat to Kazakhstan as an import-dependent country, resulting in declining living standards, inflationary pressures, and reduced purchasing power due to anti-Russian sanctions and retaliatory measures by the Russian government. Central Asian countries, including Kazakhstan, maintain long-term trade ties with Ukraine, exposing them to secondary impacts such as rising food prices, which generally have a negative impact on the countries' food affordability and food security. These issues are especially aggravated due to the instability of the national currency.

Forecast modelling has been the subject of research in various fundamental areas over the last decade, from economics (Chuku et al. 2019; Mohamed 2022; Rakha et al. 2021) to ecology (Guo et al. 2022; Munch et al. 2017; Sun et al. 2022; Kujawska et al. 2022) and natural hazards (Jain et al. 2018; Oishi et al. 2015). Nevertheless, forecasting in the context of food security is a relatively new method. Therefore, it is crucial to understand how to forecast the food security crises and suggest effective future steps using the modelling data (Alfred et al. 2022; Christensen et al. 2021; Lutoslawski et al. 2021). Prediction models for total food consumption and each type of food product present a demanding endeavor, particularly due to the multitude of socioeconomic factors influencing them.

Existing studies in Kazakhstan, in terms of theoretical methodology, include only the emergy method (Jia and Zhen 2021), the scenario approach (Khishauyeva et al. 2017), and

the method of integrated indicators (Stukach et al. 2022). The new approach proposed in this article relies on neural network modelling, specifically tailored to the unique context of Kazakhstan's food consumption patterns, thereby enhancing our theoretical understanding of how socioeconomic factors intersect with food security issues. Other researchers have explored various combinations of modelling and architecture models to address similar challenges. A neural network is a powerful tool for identifying complex trends within datasets. Abdella et al. (2020) leveraged the impact of environmental, economic, and social indicators on the 29 food consumption categories in the USA using k-means clustering and logistic regression, which are machine learning methods. The model's results showed that the overall accuracy was 91.67%, indicating considerable efficiency. In a different work, Wang et al. (2010) used GM (1, 1) grey modelling, BP neural network modelling, and a composite of grey–neural network modelling to predict global food consumption. The best result was demonstrated by the combination model, considering efficiency based on M (mean absolute error), MP (average relative error), and T (Theil inequality coefficient). Gerber Machado et al. (2020) employed a neural network based on four independent variables—sex, ethnicity, level of education, and income group (input layer)—with four neurons in the hidden layer, predicting the dependent variable of food consumption per household per capita per day in Brazil (output layer). The value of the regression coefficient R at the level of 0.90958 indicates significant network efficiency. Alfred et al. (2022) applied machine learning methods to model the consumption per capita of 33 fresh agro-food items in Malaysia based on the total GDP per capita. A resilient backpropagation neural network was applied in their research. The neural network included three types of models, where the first model, with one input, one hidden layer (10 neurons), and one output, demonstrated the lowest total MSE (mean square error) of 17.95 compared to the other types of models. Considering the consistent evidence from various studies, it is clear that neural networks exhibit high-quality forecasting capabilities. Hence, we utilized them in our research endeavors. Moreover, it is essential to note that these neural networks differ in structure, parameters, and methodologies. Each research problem has its unique characteristics and intricacies. Therefore, it is imperative to treat each research challenge individually, customizing the neural network's architecture and parameters to best suit the specific dataset and objectives at hand.

Understanding the relationship between factors and food consumption is crucial for formulating effective strategies to promote food security. The recognition of the influence characteristics is a precondition to identifying the risks that they pose and implementing countermeasures. The increase in the population can intensify the pressure on food consumption (Jia et al. 2023). Household sizes, due to household economies of scale, decrease the food consumption per capita (Nelson 1988), while household income influences consumption expenditure patterns (Barigozzi et al. 2012; Tajaddini and Gholipour 2018). The unemployment rate affects household budgets and, as a result, changes the dietary intake (Antelo et al. 2017; Smed et al. 2018) and influences dietary composition and food consumption patterns. Economic factors such as gross domestic product (GDP) (indicating the economic prosperity of a certain country) (Jia et al. 2023; Martini et al. 2022) determine the income level and its increase, especially in less developed countries, which can result in increased food demand and changes in its structure (lower income influences the food choice) (Erokhin et al. 2021). As mainly low-income social groups are susceptible to the risk of food insecurity, the income distribution matters (Rosen and Shapouri 2001; Rashidi Chegini et al. 2021). Reducing income inequalities can be expected to contribute to a decline in the risk of food insecurity. The rise in food price inflation negatively affects the food consumption models (Bozsik et al. 2022; Green et al. 2013; Martini et al. 2022). An increase in the poverty rate means a drop in food purchasing power, intensifies threats of undernutrition (thereby increasing the risk of food insecurity) (Hjelm et al. 2016; Siddiqui et al. 2020), and reduces people's food options, impacting the adjustment in the food consumption structure (Chang et al. 2009). The average subsistence level per capita to some extent corresponds to the poverty rate. Its increase means that consumers have to change their

consumption structure. According to Jensen and Miller (2008), the elasticity of demand (food) is substantially dependent on the severity of poverty.

Predicting and mitigating the risk of food insecurity is a vital issue amid geopolitical conflicts, pandemics, and economic crises. Accurate predictions of food consumption trends allow governments to allocate resources to the risk-prone areas where shocks affect vulnerable populations. Moreover, government organizations can leverage the findings to formulate targeted interventions and develop strategies that mitigate the impact of shocks on food security. This study aligns with the objectives outlined in the "The National Plan for Ensuring Food Security of the Republic of Kazakhstan for the Years 2022–2024", providing valuable insights to the government for addressing current and emerging food security challenges.

The main goal of this paper is to develop a neural network predictive model for forecasting food consumption patterns in Kazakhstan, so as to mitigate the risk of food insecurity. The first step of this research was the identification of socioeconomic variables that have a significant impact on food consumption patterns in Kazakhstan. For this purpose, principal component analysis (PCA) formed the basis for the subsequent analysis and made it possible to determine the most critical variables.

Taking into account the analysis of the literature on socioeconomic factors affecting food consumption as input data for the PCA analysis, the following variables were used: population growth rate (%), GDP per capita (KZT), food price and tariff index (previous year = 100), poverty rate (%), income concentration ratio (Gini index), average household size (people), average subsistence level per capita (KZT), and unemployment rate (%).

The main research contributions of this paper are as follows:

- Analyzing historical data related to food consumption and economic factors, and identifying dynamic trends and patterns.
- Using PCA analysis to indicate the key variables/factors for food consumption and their values for modelling a neural network.
- Demonstrating the computational power of an artificial neural network (ANN) to create models capable of successfully predicting total food consumption and the percentage distribution of different food consumption categories.
- Establishing the foundational architecture and parameters of the ANN models.
- Creating a neural network model that can be used to directly predict total food consumption and the percentage distribution of different food consumption categories, thus presenting an alternative to the existing calculation methods. Thus, the government has another useful method to develop economic policies tailored to current and short-term needs.

## 2. Materials and Methods

### 2.1. Datasets

The Bureau of National Statistics of the Republic of Kazakhstan provides an estimation of food consumption using the primary ten types of products: bakery products and cereals, meat and meat products, fish and seafood, milk and dairy products, eggs, oils and fats, fruits, vegetables, potatoes, and sugar, jam, honey, chocolate, and confectionery as the final category. In order to build a robust forecasting model, six prominent positions that mainly influence the daily nutrition of a typical person were selected: bakery products and cereals, meat and meat products, milk and dairy products, oils and fats, fruits and vegetables, and potatoes. The data of the Bureau of National Statistics of the Republic of Kazakhstan (2023) show that the total food consumption per capita per month witnessed some fluctuations between 2012 and 2020; it increased from 69.1 kg to 81.1 kg, and then there was a downward trend to 74.5 kg in 2022.

In order to predict the level of food consumption in the Republic of Kazakhstan using ANN modelling, the study extracted historical data for 22 years from the Bureau of National Statistics. The data containing the economic factors used for the conducted research are

presented in Table 1. Data indicating the levels of food consumption for six food product categories are presented in Table 2.

**Table 1.** Analyzed economic factors.

| Year | Population Growth Rate | GDP per Capita | Food Price and Tariff Index | Poverty Rate | Income Concentration Ratio | Average Household Size | Average Subsistence Level per Capita | Unemployment Rate |
|------|------|------|------|------|------|------|------|------|
| | (%) | (KZT) | (Previous Year = 100) | (%) | (Gini Index) | (People) | (KZT) | (%) |
| 2000 | −0.36 | 311,409 | 113.9 | 31.8 | 0.307 | 3.4 | 4007 | 12.8 |
| 2001 | −0.24 | 354,051 | 110.0 | 46.7 | 0.366 | 3.7 | 4596 | 10.4 |
| 2002 | −0.10 | 388,732 | 108.1 | 44.5 | 0.328 | 3.6 | 4761 | 9.3 |
| 2003 | 0.11 | 423,457 | 108.2 | 37.5 | 0.315 | 3.6 | 5128 | 8.8 |
| 2004 | 0.57 | 460,895 | 108.3 | 33.9 | 0.305 | 3.5 | 5427 | 8.4 |
| 2005 | 0.83 | 501,128 | 109.0 | 31.6 | 0.304 | 3.5 | 6014 | 8.1 |
| 2006 | 0.96 | 548,912 | 107.7 | 18.2 | 0.312 | 3.4 | 8410 | 7.8 |
| 2007 | 1.17 | 590,966 | 129.3 | 12.7 | 0.309 | 3.4 | 9653 | 7.3 |
| 2008 | 1.13 | 599,141 | 111.2 | 12.1 | 0.288 | 3.3 | 12,364 | 6.6 |
| 2009 | 2.64 | 594,429 | 101.2 | 8.2 | 0.267 | 3.4 | 12,660 | 6.6 |
| 2010 | 1.38 | 628,871 | 110.2 | 6.5 | 0.278 | 3.4 | 13,487 | 5.8 |
| 2011 | 1.46 | 665,808 | 109.8 | 5.5 | 0.290 | 3.5 | 16,072 | 5.4 |
| 2012 | 1.42 | 688,007 | 105.1 | 3.8 | 0.284 | 3.5 | 16,815 | 5.3 |
| 2013 | 1.42 | 718,864 | 102.7 | 2.9 | 0.276 | 3.4 | 17,789 | 5.2 |
| 2014 | 1.48 | 738,106 | 107.3 | 2.9 | 0.278 | 3.4 | 19,068 | 5.0 |
| 2015 | 1.49 | 736,126 | 110.1 | 2.6 | 0.278 | 3.4 | 19,647 | 5.1 |
| 2016 | 1.46 | 733,715 | 108.9 | 2.5 | 0.278 | 3.4 | 21,612 | 5.0 |
| 2017 | 1.41 | 753,478 | 106.2 | 2.7 | 0.287 | 3.4 | 23,783 | 4.9 |
| 2018 | 1.33 | 774,127 | 104.8 | 4.3 | 0.289 | 3.4 | 27,072 | 4.9 |
| 2019 | 1.31 | 798,597 | 109.6 | 4.3 | 0.290 | 3.4 | 29,342 | 4.8 |
| 2020 | 1.28 | 768,586 | 111.4 | 5.3 | 0.291 | 3.4 | 33,015 | 4.9 |
| 2021 | 1.33 | 791,285 | 110.0 | 5.2 | 0.294 | 3.4 | 37,579 | 4.9 |
| 2022 | 3.30 | 790,763 | 125.5 | 5.2 | 0.285 | 3.4 | 43,566 | 4.9 |

During the first three years of the analyzed period, the population of Kazakhstan decreased, but in 2003 the trend reversed (Table 1). From 2007 onward, the dynamics remained relatively constant. Two years stood out with significantly high dynamics. Both cases were due to high migration. In 2009, the economic crises following the 2008 financial crisis were the reason, while in 2022, the Ukrainian–Russian war played a role (Agaidarov et al. 2023). Since the 2000s, Kazakhstan has experienced impressive economic growth (World Bank 2023). The GDP per capita more than doubled during the examined period; however, the growth rate fluctuated and, in some years, was negative. Food prices were rising rapidly, and only in a few years where the dynamics were lower than 105. In 2022, following the Russian invasion of Ukraine, inflation soared to 25.5%. Kazakhstan has made substantial progress in reducing poverty (Kohl et al. 2005). The poverty rate was initially high but decreased until 2016, after which it increased and stabilized at 5.2%. The trend in the average household size showed a decline. In 2001, the average household size was 3.7 people, while in 2022 it was 3.4. There was a noticeable reduction in the income concentration ratio during the examined period, suggesting a more equal distribution of

national income. The average subsistence level per capita, expressed in KZT, rose gradually, and in some years quite rapidly, partially due to the rise in food prices. The situation in the labor market improved gradually, and the unemployment rate fell by more than half, remaining below 5% since 2017.

**Table 2.** Total food consumption and food consumption for 6 food product categories.

| Year | Total Food Consumption (per Capita per Month in kg) | Bakery Products and Cereals (kg) | (%) | Meat and Meat Products (kg) | (%) | Milk and Dairy Products (kg) | (%) | Oils and Fats (kg) | (%) | Fruits and Vegetables (kg) | (%) | Potatoes (kg) | (%) |
|---|---|---|---|---|---|---|---|---|---|---|---|---|---|
| 2000 | 59.20 | 10.30 | 17.40 | 3.7 | 6.25 | 19.6 | 33.11 | 0.9 | 1.52 | 8.5 | 14.36 | 5.5 | 9.29 |
| 2001 | 63.10 | 10.00 | 15.85 | 3.7 | 5.86 | 19.6 | 31.06 | 1.5 | 2.38 | 10.8 | 17.12 | 5.5 | 8.72 |
| 2002 | 61.90 | 10.00 | 16.16 | 3.8 | 6.14 | 19.3 | 31.18 | 1.2 | 1.94 | 9.5 | 15.35 | 5.4 | 8.72 |
| 2003 | 58.60 | 10.20 | 17.41 | 3.4 | 5.80 | 16.7 | 28.50 | 1.1 | 1.88 | 9.3 | 15.87 | 4.6 | 7.85 |
| 2004 | 54.60 | 9.70 | 17.77 | 3.3 | 6.04 | 15.8 | 28.94 | 0.9 | 1.65 | 8.9 | 16.30 | 4.1 | 7.51 |
| 2005 | 54.40 | 9.50 | 17.46 | 3.3 | 6.07 | 15.8 | 29.04 | 0.9 | 1.65 | 8.9 | 16.36 | 3.9 | 7.17 |
| 2006 | 57.90 | 10.30 | 17.79 | 3.7 | 6.39 | 17.1 | 29.53 | 0.9 | 1.55 | 9.4 | 16.23 | 3.8 | 6.56 |
| 2007 | 59.50 | 10.20 | 17.14 | 4.1 | 6.89 | 17.3 | 29.08 | 0.9 | 1.51 | 9.7 | 16.30 | 3.8 | 6.39 |
| 2008 | 58.80 | 10.20 | 17.35 | 4.1 | 6.97 | 17.0 | 28.91 | 0.9 | 1.53 | 9.7 | 16.50 | 3.7 | 6.29 |
| 2009 | 60.70 | 10.10 | 16.64 | 4.2 | 6.92 | 17.5 | 28.83 | 1.1 | 1.81 | 10.2 | 16.80 | 3.6 | 5.93 |
| 2010 | 60.20 | 10.20 | 16.94 | 4.4 | 7.31 | 17.0 | 28.24 | 1.1 | 1.83 | 9.9 | 16.45 | 3.5 | 5.81 |
| 2011 | 69.30 | 10.40 | 15.01 | 5.5 | 7.94 | 19.0 | 27.42 | 1.6 | 2.31 | 12.2 | 17.60 | 4.0 | 5.77 |
| 2012 | 69.10 | 10.30 | 14.91 | 5.6 | 8.10 | 18.4 | 26.63 | 1.5 | 2.17 | 12.1 | 17.51 | 4.1 | 5.93 |
| 2013 | 70.60 | 10.40 | 14.73 | 5.8 | 8.22 | 19.0 | 26.91 | 1.5 | 2.12 | 12.4 | 17.56 | 4.1 | 5.81 |
| 2014 | 70.40 | 10.52 | 14.93 | 5.9 | 8.36 | 18.8 | 26.70 | 1.6 | 2.21 | 12.3 | 17.45 | 4.0 | 5.74 |
| 2015 | 73.00 | 10.80 | 14.79 | 6.1 | 8.36 | 19.5 | 26.71 | 1.6 | 2.19 | 12.9 | 17.67 | 4.0 | 5.48 |
| 2016 | 72.70 | 10.90 | 14.99 | 6.1 | 8.39 | 19.6 | 26.96 | 1.6 | 2.20 | 12.5 | 17.19 | 4.0 | 5.50 |
| 2017 | 73.70 | 11.14 | 15.12 | 6.1 | 8.25 | 19.8 | 26.88 | 1.6 | 2.21 | 12.8 | 17.31 | 3.9 | 5.30 |
| 2018 | 80.60 | 11.54 | 14.32 | 6.5 | 8.05 | 21.8 | 27.01 | 1.6 | 1.98 | 14.1 | 17.47 | 4.0 | 5.02 |
| 2019 | 79.10 | 11.40 | 14.41 | 6.6 | 8.34 | 21.1 | 26.68 | 1.4 | 1.77 | 13.6 | 17.19 | 4.0 | 5.06 |
| 2020 | 81.10 | 11.69 | 14.42 | 7.0 | 8.60 | 21.6 | 26.66 | 1.4 | 1.78 | 13.8 | 16.96 | 4.2 | 5.15 |
| 2021 | 77.80 | 11.10 | 14.27 | 6.9 | 8.87 | 20.3 | 26.09 | 1.4 | 1.80 | 13.1 | 16.84 | 3.9 | 5.01 |
| 2022 | 74.50 | 10.70 | 14.36 | 6.5 | 8.72 | 18.9 | 25.37 | 1.3 | 1.74 | 12.6 | 16.91 | 3.7 | 4.97 |

The data in Table 2 show that during the first two decades of the 21st century, food consumption per capita in Kazakhstan considerably increased, from 59 kg in 2000 to 81 kg in 2020. However, in the following two years, there was a reduction of 6 kg. The trends in the consumption of the different product groups differed. Although the consumption of bakery and milk products increased slightly, their shares in total food consumption decreased. Fruits and vegetables attracted more attention from consumers, but their role in the food basket remained relatively stable. The consumption of oils rose in the first decade and then stabilized; as a result, their share in consumption fell. Meat consumption soared, nearly doubling, so the share of meat increased markedly. The increase in the consumption of fruits and meat (Jia et al. 2022) is connected with societal and economic development. Potato consumption dropped by 33%, resulting in a twofold reduction in its share in the food basket.

### 2.2. Neural Network

In the initial phase of the investigation, the task was to identify the contributory factors necessary for the ANN modelling. To achieve this end, the method of PCA was employed within the framework of the Statistica 13.1 milieu (TIBCO Software Inc., Palo Alto, CA, USA). This technique, a widely recognized stratagem for the contraction of dimensionality in multifaceted data scrutiny and statistical inquiries, endeavors to transmute the primordial, extensive data into a novel sequence of orthogonal variables denominated as principal components. The goal is to preserve as much of the variance in the data as possible. These principal components are arranged so that the first component captures the largest variance in the data, the second component captures the second-largest variance, and so on.

The eigenvalues and eigenvectors of the covariance matrix are determined during a PCA. These eigenvectors symbolize the principal components, while their corresponding eigenvalues provide a quantifiable description of the amount of variance that is explained by each component. A judicious selection of principal components is then made. Those with the highest eigenvalues are considered to be the principal components. In the studies reviewed, it was assumed that a set of components would be selected equal to that which is indispensable to explain approximately a bare minimum of 95% of the dispersion in the data.

The following economic factors were used as input parameters for the PCA analysis: population growth rate (%), GDP per capita (KZT), food price and tariff index (previous year = 100), poverty rate (%), income concentration ratio (Gini index), average household size (people), average subsistence level per capita (KZT), and unemployment rate (%). The input parameters and their values for modelling the neural network were selected based on the PCA analysis.

In the next stage of this research, the Neural Net Fitting neural network library of the MATLAB software (https://uk.mathworks.com/products/matlab.html (accessed on 15 September 2023)) package was used to build the ANN models. Two different models were developed during this research: The first model was designed to predict food consumption on a national scale per capita per month for the next three years. The second model concerned forecasting the percentage distribution of various food consumption categories, also for the next three years. The categories selected for this modelling exercise included bakery products and cereals (%), meat and meat products (%), milk and dairy products (%), oils and fats (%), fruits and vegetables (%), and potatoes (%).

The employment of a shallow neural network was undertaken for the training process, utilizing the Levenberg–Marquardt algorithm as the primary learning algorithm. The resulting networks consisted of a single hidden layer. The number of neurons within this hidden layer, ranging from 2 to 10, was determined through an experimental approach. A diagrammatic representation of the artificial neural network is depicted in Figure 1. To establish the connections, 20 parameter sets were utilized, which were subsequently divided into training data (accounting for 80%) and validation data (comprising the remaining 20%). Due to the small number of modelling data, test data were omitted.

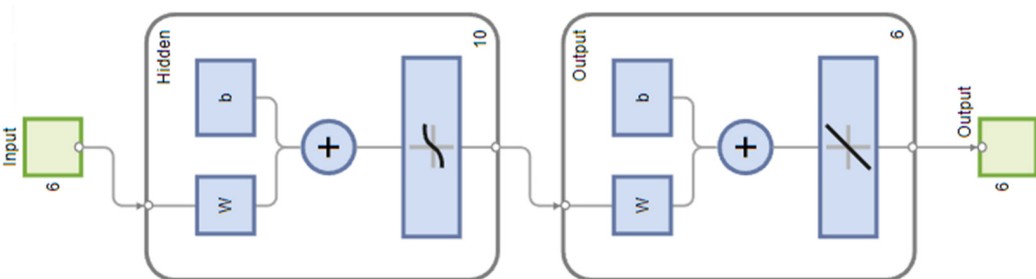

**Figure 1.** A schematic representation of an artificial neural network.

The determination of the optimal network was predicated on the evaluation of the network's quality indices, which encompassed the mean square error (MSE), the mean

absolute error (MAE), and the regression (R) value. The computation of the MSE was achieved through the following equation:

$$\text{MSE} = \frac{1}{n}\sum\nolimits_{i=1}^{n}\left(y_i - y_i'\right)^2 \tag{1}$$

where n is the total number of instances within a given set, $y_i$ represent the real data for the i-th observation, and $y_i'$ is the predicted value of the data for the i-th observation.

The mean absolute error (MAE) is a statistical measure computed as the average of the absolute differences between the predicted and actual values, using the following formula:

$$\text{MAE} = \frac{1}{n}\sum\nolimits_{i=1}^{n}\left(\left|y_i - y_i'\right|\right) \tag{2}$$

The regression value R quantifies the correlation between the outputs and the inputs. It provides an indication of the degree to which the predicted outputs match the actual outputs: a well-trained network is characterized by an R value that approximates 1. The calculation of the regression R values was conducted in accordance with the following equation:

$$R\left(y', y^*\right) = \frac{\text{cov}(y, y')}{\sigma_{y'}\sigma_y}R\epsilon < 0, 1 > \tag{3}$$

where $\sigma_y$ is the standard deviation of the real data, while $\sigma_{y'}$ is the standard deviation of the predicted data.

The quality of the generated network can be improved by increasing the regression coefficient R and decreasing the MSE and MAE.

## 3. Results

### 3.1. Selection of Input Parameters for Neural Network Modelling

PCA was employed in the selection of parameters for neural network modelling. The eigenvalues of the principal components were computed, and their quantity was initially restricted to eight by the software. As discernible from Table 3 and the information delineated in Figure 2, the first four principal components (marked red) exert the predominant influence. The cumulative percentage of variance for the first four principal components yields 95.66% of the variance of the principal components, signifying that these four primary elements encompass 95% of the information (as measured by variance) that the original eight characteristics contained.

**Table 3.** The cumulative % of variance for the principal components.

| ID | Eigenvalues | % of Total Variance | Cumulative Eigenvalues | Cumulative % |
|----|-------------|---------------------|------------------------|--------------|
| 1 | 5.416777 | 67.70972 | 5.416777 | 67.7097 |
| 2 | 1.137102 | 14.21378 | 6.553880 | 81.9235 |
| 3 | 0.756293 | 9.45366 | 7.310173 | 91.3772 |
| 4 | 0.342751 | 4.28438 | 7.652923 | 95.6615 |
| 5 | 0.198354 | 2.47942 | 7.851277 | 98.1410 |
| 6 | 0.090914 | 1.13642 | 7.942191 | 99.2774 |
| 7 | 0.052033 | 0.65042 | 7.994224 | 99.9278 |
| 8 | 0.005776 | 0.07220 | 8.000000 | 100.0000 |

In order to identify the attributes that have the most substantial influence on each principal component, we can examine the characteristics of the PCA components. This characteristic within the PCA object provides us with the coefficients related to each attribute within the principal components. The higher the absolute value of the coefficient, the greater the influence of that attribute on the principal component. Table 4 shows the variables and their respective influences on the principal components, with the two

variables that have the most significant influence on the principal components highlighted in red.

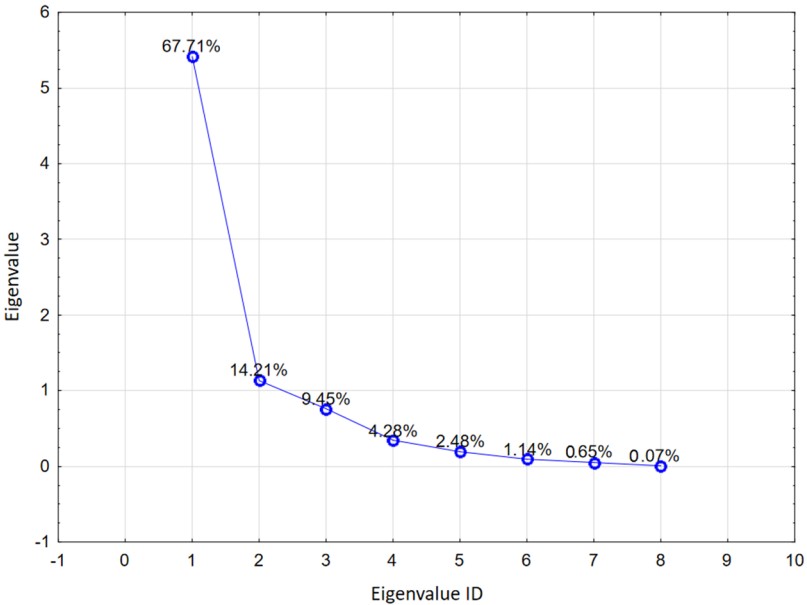

**Figure 2.** Scree plot of eigenvalues for PCA showing the percentage of variance explained by each component.

**Table 4.** Factor coordinates of variables.

| Variables | Factor Coordinates of Variables | | | |
|---|---|---|---|---|
| | Principal Component #1 | Principal Component #2 | Principal Component #3 | Principal Component #4 |
| Population growth rate (%) | −0.864404 | 0.124719 | −0.019662 | 0.452331 |
| GDP per capita (KZT) | −0.957218 | 0.000139 | −0.233246 | −0.132317 |
| Food price and tariff index (previous year = 100) | −0.012964 | 0.981131 | 0.118890 | 0.046963 |
| Poverty rate (%) | 0.964009 | 0.062631 | −0.075177 | 0.065354 |
| Income concentration ratio (Gini index) | 0.860438 | 0.263823 | −0.341785 | −0.145397 |
| Average household size (people) | 0.752328 | −0.081289 | −0.584805 | 0.241730 |
| Average subsistence level per capita (KZT) | −0.817264 | 0.276860 | −0.382275 | −0.185840 |
| Unemployment rate (%) | 0.921718 | 0.113222 | 0.277069 | −0.007256 |

After the analysis, the following variables were selected for further study via neural network modelling:

- Natural population growth (%);
- GDP per capita (KZT);
- Food price and tariff index (previous year = 100);
- Poverty rate (%);
- Average household size (people);
- Average subsistence level per capita (KZT).

The values of principal components #1 and #2, connected to the above variables, indicate that the poverty rate, GDP per capita, and food price index are the most important factors for food consumption in Kazakhstan.

### 3.2. Modelling the Total Food Consumption

The best modelling results were achieved using a network composed of nine neurons—a finding that was determined after 17 epochs. The results of the training for this network are outlined in Table 5.

**Table 5.** The results of learning for the best neural network and the quality indicators of this network.

| Training Algorithm | Levenberg–Marquardt |
|---|---|
| Epoch | 7 |
| Performance | $4.42 \times 10^{-29}$ |
| Best validation performance | 4.3324 at epoch 4 |
| Gradient | $1.19 \times 10^{-13}$ |
| R (all) | 0.99488 |
| MSE | 0.8679 |
| MAE | 0.3435 |

The optimal validation was reached at epoch four, as depicted in Figure 3. Figure 4 includes regression plots for training, validation, and all datasets.

Forecasts of the total food consumption are presented in Figure 5. For comparison, the data obtained from modelling are collated alongside the actual data; additionally, predictions for the next three years are also shown. The results suggest a decrease in food consumption in Kazakhstan over three consecutive years. The rapid population increase connected to the Ukrainian–Russian war migration and the high inflation are the main reasons for this phenomenon.

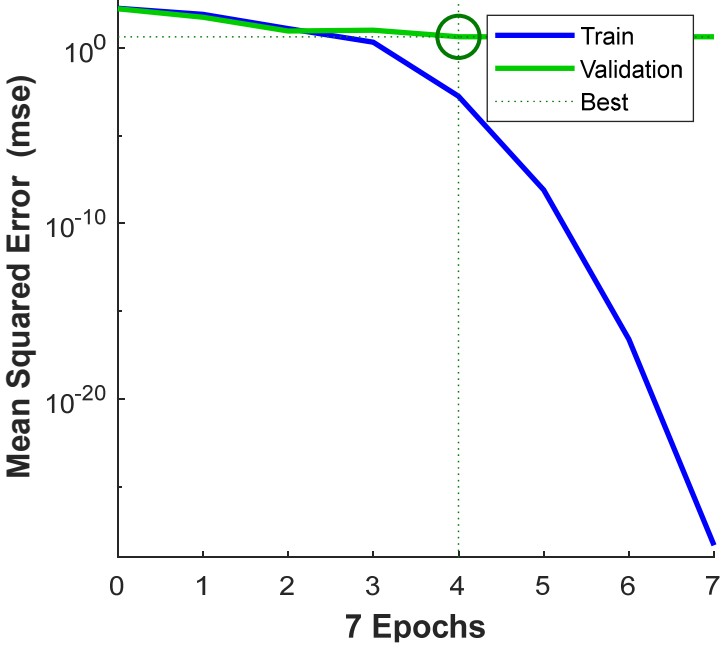

**Figure 3.** The training process of the artificial neural network for the total food consumption model (the best validation performance is marked by a green circle).

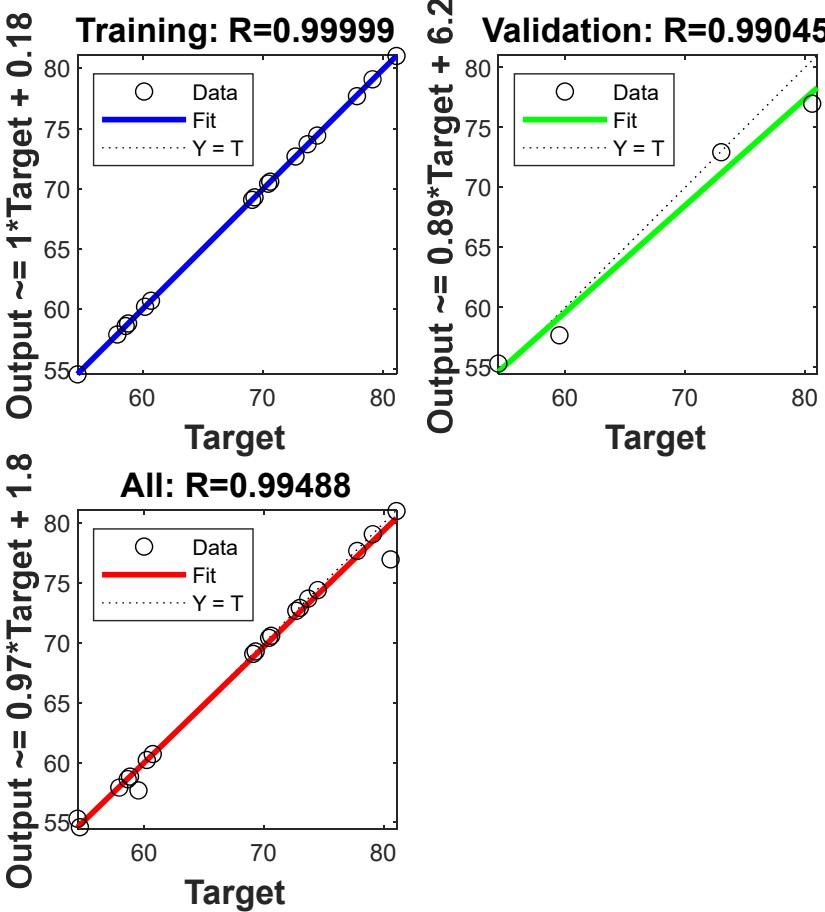

**Figure 4.** Three regression plots (statistics for individual sets and the total set) correlating the targets with the outputs of the neural network for the total food consumption model.

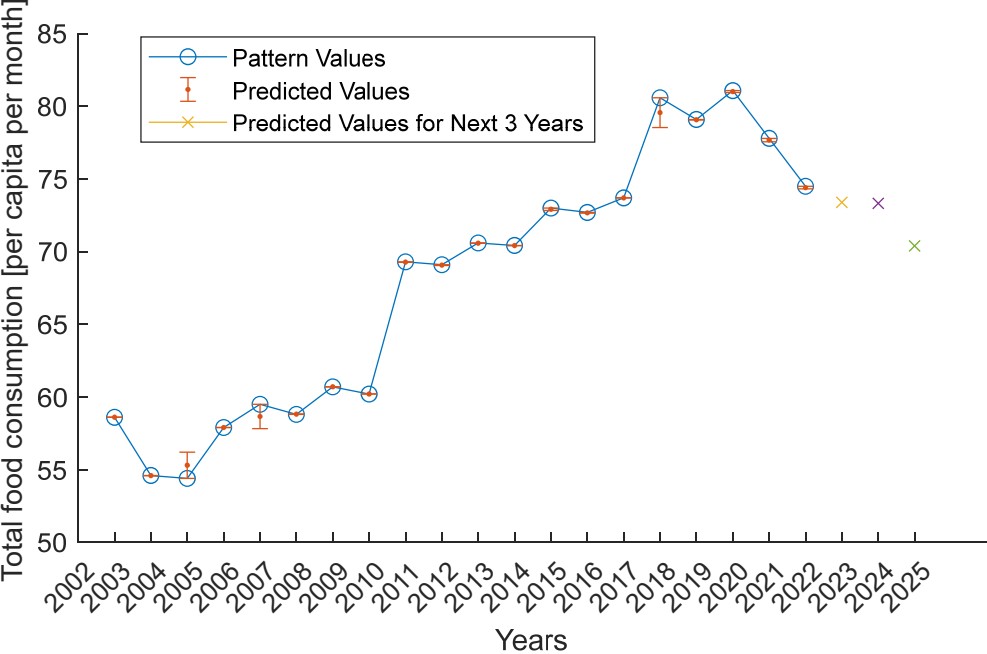

**Figure 5.** Forecasts of the total food consumption from 2003 to 2025.

The prediction results exhibit low prediction error, below 5%. In the aggregate, the positive modelling quality indicators strengthen the assumption that network modelling can be used to predict the total food consumption (per capita per month) in Kazakhstan.

### 3.3. Modelling the Percentage Distribution of Various Food Consumption Categories

The most optimal modelling outcomes were achieved using a network composed of eight neurons—a result that was ascertained after 11 epochs. The outcomes of the training for this network are delineated in Table 6. Training halted following six successive rises in the validation error (or absences of error reduction), and the optimal outcomes were derived from the iteration with the least validation error.

**Table 6.** The results of learning for the best neural network and the quality indicators of this network.

| Training Algorithm | Levenberg–Marquardt |
|---|---|
| Epoch | 11 |
| Performance | 0.00479 |
| Best validation performance | 0.14962 at epoch 5 |
| Gradient | 0.00629 |
| R (all) | 0.99973 |
| MSE | 0.0392 |
| MAE | 0.1239 |

The quality efficacy of the ANNs was scrutinized utilizing the MSE metric. The results indicated that the optimal validation was achieved during the fifth epoch, as depicted in Figure 6, and the error histogram distribution is shown in Figure 7. Given that the histogram's shape mirrors a Gaussian distribution curve, and that the majority of errors are of the lowest values, we can infer that the trained network is of high quality, with no signs of overfitting. Figure 8 includes regression plots for training (R = 0.99992), validation (R = 0.99899), and all datasets (R = 0.99973).

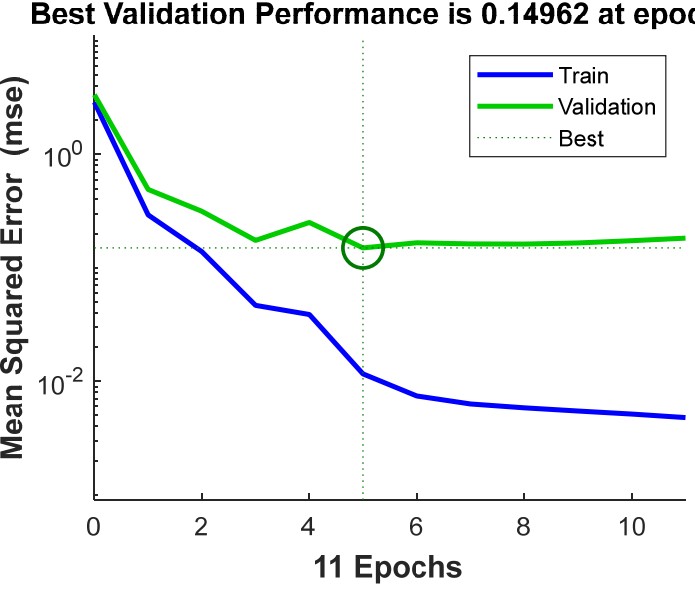

**Figure 6.** The ANNs' training performance over 11 epochs for food consumption category distribution modelling (the best validation performance is marked by a green circle).

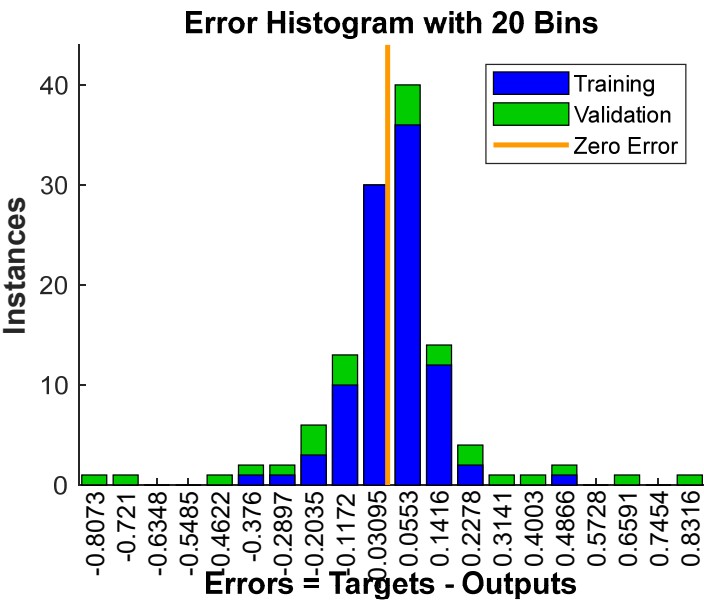

**Figure 7.** Error histogram distribution—the frequency distribution of the prediction errors for both the training and validation sets.

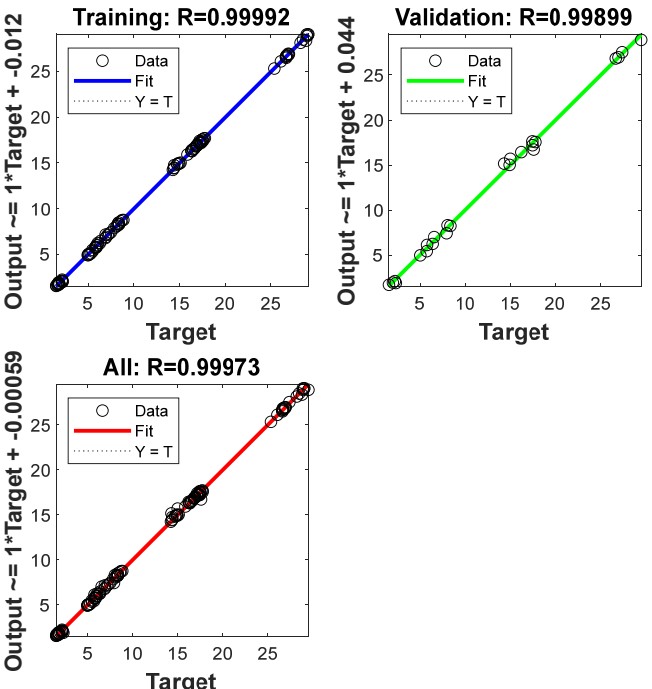

**Figure 8.** Regression statistics for the ANN model: actual vs. predicted values for training and validation sets, with R-values.

Forecasts of the percentage distribution of the various food consumption categories according to the established categories—bakery products and cereals, meat and meat products, milk and dairy products, oils and fats, fruits and vegetables, and potatoes—are presented in Figure 9. For comparison, data obtained from modelling and actual data were collated; additionally, predictions were made for the next three years. The model shows that in 2023 and 2024, the trends in the food consumption structure will continue: the shares of bakery products will remain nearly unchanged, the participation of meat, potatoes, and fruits and vegetables will rise, and that of dairy products and oils will drop. The most

noticeable changes will occur in 2025, with a sharp decline in the share of potatoes and meat and a surge in the importance of bakery and dairy products.

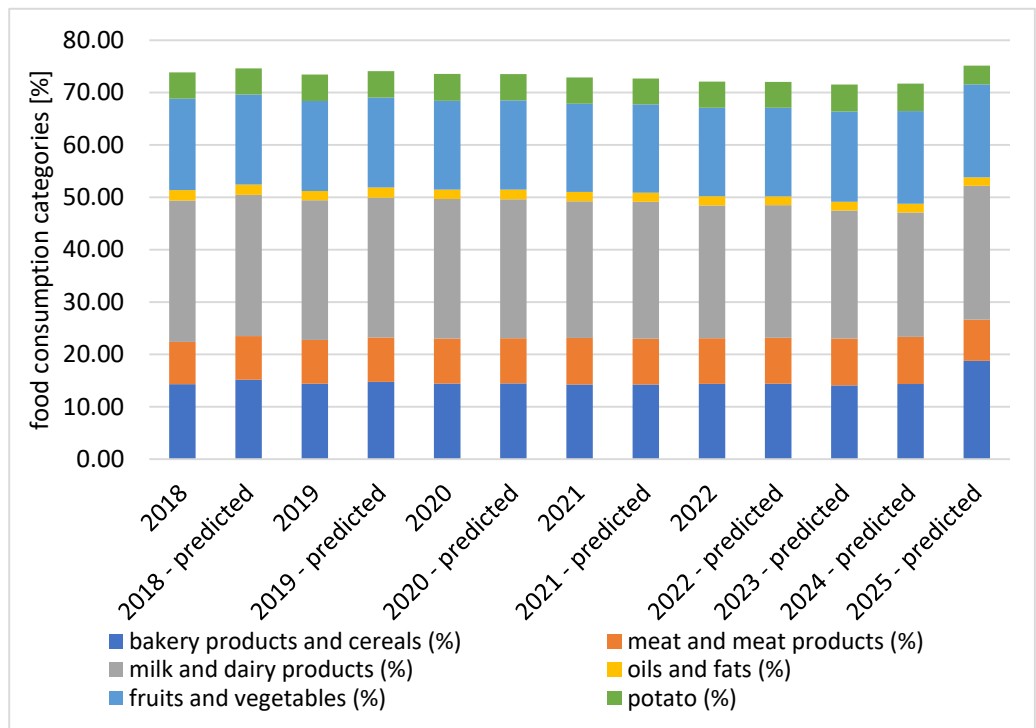

**Figure 9.** Forecasts of the percentage distribution of various food consumption categories from 2018 to 2025.

The prediction results exhibit low prediction error, below 10%, as confirmed by the regression value (R = 0.99973). In the aggregate, the positive modelling quality indicators strengthen the assumption that network modelling can be used to predict the percentage distribution of various food consumption categories in Kazakhstan.

## 4. Discussion

The scientific investigations outlined in this research provide a comprehensive understanding of the use of different machine learning approaches to predict food consumption patterns. These investigations highlight the potential of these methods for accurate forecasts, which can be valuable for shaping policies and strategies for food security.

This study is notable for employing neural networks to predict how various food consumption categories are distributed in Kazakhstan. Moreover, the investigation revealed that the best setup for distributing the various food consumption categories and the overall food consumption involved using a network with eight neurons and nine neurons, respectively. The minimal prediction error and high regression value both for the model for the proportional distribution of various food consumption categories (R = 0.99973) and for the total food consumption model (R = 0.99488) demonstrate the effectiveness of the neural network model in forecasting food consumption trends. High R values approaching 1 indicate a strong correlation between the predicted values and actual observations, emphasizing the robust performance of the model in capturing and predicting complex food consumption patterns.

These findings suggest that neural network modelling could be a valuable tool for predicting food consumption trends, assisting in planning and policymaking for food security.

Examining global research, the investigation into the per capita consumption of several fresh agro-food commodities in Malaysia highlights the effectiveness of neural networks. The study showed that the neural network (NN) model outperformed the ordinary least

squares (OLS) model, resulting in a lower aggregate mean squared error (MSE). The NN model demonstrated the lowest total MSE of 17.95 for all 33 fresh agro-foods investigated in this study (Alfred et al. 2022). This finding further strengthens the argument for using neural networks in forecasting food consumption trends.

Studies by Giulia Martini and associates, Pietro Foini et al., and Deléglise et al. highlight the application of machine learning in predicting food security situations. Martini et al. (2022) employed XGBoost, explaining up to 81% of the variation in insufficient food consumption and up to 73% of the variation in crisis or above food-based coping levels. The research by Foini et al. (2023) demonstrated that precise forecasts of insufficient food consumption levels could be made up to 30 days into the future, thereby informing decisions regarding the allocation of need-based humanitarian assistance. However, Deléglise et al. (2020) discovered that predicting food security indices is a challenging issue; their models did not exceed $R^2 = 0.38$ for the Household Dietary Diversity Score (HDDS) and $R^2 = 0.35$ for the Food Consumption Score (FCS).

These findings collectively show the potential of machine learning paradigms, especially neural networks, for predicting food consumption trends. While the successful application of neural networks in forecasting consumption patterns is evident, this research lays a strong foundation for future explorations in this field. The ability of machine learning models, such as neural networks, to clarify differences in food consumption and coping levels opens promising opportunities for further research and progress. These insights contribute to the increasing knowledge about using advanced computational methods to better understand the dynamics of food security.

## 5. Conclusions

This paper presents a comprehensive study focused on building a neural network model to forecast food consumption trends in Kazakhstan, with the goal of reducing the potential risk of food insecurity. The initial phase of the research concentrated on identifying socioeconomic factors that significantly influence how people consume food in Kazakhstan. To achieve this, we used principal component analysis (PCA) as the main method for further evaluations. PCA facilitated the identification of important factors, such as natural population growth (%), GDP per capita (KZT), food price and tariff index (previous year = 100), poverty rate (%), average household size (people), and average subsistence level per capita (KZT). In addition, considering the PCA results, especially the values of principal components #1 and #2, revealed that the most crucial factors affecting food consumption in Kazakhstan are the poverty rate, GDP per capita, and food price index.

The results of the PCA analysis were the input variables used to build the ANN models. Two models were built: to predict food consumption on a national scale per capita per month, and to model the percentage distribution of various food consumption categories. The general regression for the first network was 0.99488, while that for the second was 0.9973.

The forecast results show a low prediction error of less than 10%, signifying a high level of accuracy in the model's predictions. A low prediction error is crucial in forecasting, as it indicates that the model's estimates closely align with the actual values. Overall, the positive indicators in modelling quality support the idea that network modelling can predict total food consumption (per capita per month) in Kazakhstan and the percentage distribution of different food consumption categories in Kazakhstan. Examining the forecasted total food consumption for the next three years suggests a decrease, influenced by rapid population growth due to Ukrainian–Russian war migration and high inflation. Considering the projected trends in the food consumption structure for the next two years, we anticipate the following: firstly, the proportion of bread consumption is expected to stay relatively stable; secondly, there will be an increase in the consumption of meat, potatoes, fruits, and vegetables, along with a decrease in the consumption of dairy products and oils. Notably, significant deviations from these patterns are predicted to arise in 2025, marked

by a decline in potato and meat consumption and an increase in the consumption of bread and dairy products.

**Author Contributions:** Conceptualization, A.D. (Aigerim Duisenbekova), M.K., A.D. (Alina Danilowska), M.R. and A.G.; methodology, M.K. and A.D. (Aigerim Duisenbekova); software, M.K.; validation, A.D. (Aigerim Duisenbekova), M.K. and A.D. (Alina Danilowska); formal analysis, A.D. (Aigerim Duisenbekova), M.K.; investigation, A.D. (Aigerim Duisenbekova), M.K. and A.D. (Alina Danilowska); resources, A.D. (Aigerim Duisenbekova), M.K. and A.D. (Alina Danilowska); data curation, A.D. (Aigerim Duisenbekova); writing—original draft preparation, A.D. (Aigerim Duisenbekova), M.K. and A.D. (Alina Danilowska); writing—review and editing, A.D. (Aigerim Duisenbekova), M.K. and A.D. (Alina Danilowska); visualization, M.K.; supervision, A.D. (Aigerim Duisenbekova) and A.G. All authors have read and agreed to the published version of the manuscript.

**Funding:** This research received no external funding.

**Informed Consent Statement:** Not applicable.

**Data Availability Statement:** Data are contained within the article.

**Conflicts of Interest:** The authors declare no conflicts of interest.

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
