# Peer review of "Predicting Food Consumption to Reduce the Risk of Food Insecurity in Kazakhstan"

_economies, doi:10.3390/economies12010011_

Round 1
Reviewer 1 Report
Comments and Suggestions for Authors
In this study, the authors describe a neural network model to predict food consumption trends in Kazakhstan.
There is a growing interest in forecasting food insecurity and the study tackles a timely issue. I have some concerns regarding the quality of the presentation and the methodological approach.
1. The quality of the English language could be improved. The choice of wording is sometimes a bit unusual. For instance, the title: “to decline the risk” is not a common way of saying. It may be better the wording: “to decrease/to reduce the risk”. A more correct title would be: “Predicting food consumption to reduce the risk of food insecurity in Kazakhstan”.
2. It is not very clear why the authors use an artificial neural network as a model. Why not use something simpler, given the limited amounts of training points? A linear regression, or an ARIMA model, could perform equally better. Is there a specific need for a neural network?
3. The presentation of the figures could be improved. The caption of Figure 2 is “Plot the scatter”. I suggest replacing it with some more informative text, describing the actual content of the figure. The same applies to all figures, where captions could be improved by adding more information on the content (meaning of colors, symbols, etc.). In Figure 9, the numbers on the bars are a bit distracting.
4. Figure 5 represents the main result of the study. Would it be possible to train the model on the years 2002-2019 and see the predictions made for the years 2020, 2021, and 2022?
Showing predictions for 2025, without having shown that the model could have predicted food consumption in the past, makes the results a bit weak.
Comments on the Quality of English Language
I have already mentioned the quality of English in my report.
Author Response
Dear Sir/Madame,
First of all, we would like to thank you very much for your time you spent on reviewing our paper and express our gratitude for your kind review. We would like to thank you for your all remarks and suggestions. They gave us the opportunity to improve our paper. In the table below you can find detailed explanations and information about the changes that were made in the body of our manuscript.
Yours faithfully,
Authors

Reviewer 2 Report
Comments and Suggestions for Authors
The subject is very interesting and well analyzed. It is a very topical subject, so that the adaptation of eating behavior to current and especially future changes deserves attention.
The period chosen for the study is quite long, thus being an advantage for the results obtained.
The material is well structured and the data sources, applied methods and obtained results are mentioned. In the discussion area there are a series of explanations for the results obtained.
In the case of the results of the study, is it possible to make a forecast of food consumption to avoid a food risk in Kazakhstan by environment - rural/urban?
The conclusions are clearly expressed.
Author Response
Dear Sir/Madame,
First of all, we would like to thank you very much for your time you spent on reviewing our paper and express our gratitude for your kind review. It was very supportive to read that you found our paper interesting and well-analyzed. However, we would like to thank you for your remarks and suggestions too. Please, see the attachment.
Yours faithfully,
Authors

Reviewer 3 Report
Comments and Suggestions for Authors
Dear Authors
You made an extensive work and article that applies different tecniques.
However I like the theme, I did not appreciate how it was presented.
Let me give you some highlihgts, that in my humble opinion might be research problems in your article:
1) the authors have made a serious and a deep introduction to the theme of food insecurity. However it is not justified why doing it on this particular country (Kazakhstan). As the authors point out , there are several countries across the world that suffer from the same insecurity/scarcity. Therefore, and from a review point of view it is necessary to justify the context also.
2) it is not made clear for readers, what the authors mean by "food security". The authors have explain several plans, projects and guidelines to help it, but did not , in my point of view, conceptualise qhat do they understand by it.
3)the explanation for the use of a "neural network modelling" did not convince me as a reviewer. Why' Why not the use of regression or SEM? It is related with the forecast model that the authors intend to develop?
4) however the authors applied different methods to improve the academic knowledge of food security and consumption, the contribuitions are only in terms of forecast.
5) the discussion section and the conclusions section are very focus on the methods applied by the authors, and the therminology is very hard if a manager wants it to read it and to apply.
So in resume, my main concerns are:
- what are your contribution in terms of theoretical focus and also in terms of management (in government policies terms).
-what is your research gap?
-why kazakhstan?
Hope I could help you.
Best regards
Comments on the Quality of English LanguageI advise a proof reading of the text.
Author Response

(The authors gave the same response as above.)
